# Immunogenicity after a Third COVID-19 mRNA Booster in Solid Cancer Patients Who Previously Received the Primary Heterologous CoronaVac/ChAdOx1 Vaccine

**DOI:** 10.3390/vaccines10101613

**Published:** 2022-09-26

**Authors:** Sutima Luangdilok, Passakorn Wanchaijiraboon, Nussara Pakvisal, Thiti Susiriwatananont, Nicha Zungsontiporn, Virote Sriuranpong, Panot Sainamthip, Nungruthai Suntronwong, Preeyaporn Vichaiwattana, Nasamon Wanlapakorn, Yong Poovorawan, Nattaya Teeyapun, Suebpong Tanasanvimon

**Affiliations:** 1Department of Biochemistry, Faculty of Medicine, Chulalongkorn University and the King Chulalongkorn Memorial Hospital, Bangkok 10330, Thailand; 2Phrapokklao Cancer Center of Excellence, Phrapokklao Clinical Research Center, Phrapokklao Genomic Laboratories, Phrapokklao Hospital, Mueang District, Chanthaburi 22000, Thailand; 3Division of Medical Oncology, Department of Medicine, Faculty of Medicine, Chulalongkorn University and the King Chulalongkorn Memorial Hospital, Bangkok 10330, Thailand; 4Department of Pharmacology, Faculty of Medicine, Chulalongkorn University and the King Chulalongkorn Memorial Hospital, Bangkok 10330, Thailand; 5Center of Excellence in Clinical Virology, Department of Pediatrics, Faculty of Medicine, Chulalongkorn University and the King Chulalongkorn Memorial Hospital, Bangkok 10330, Thailand

**Keywords:** COVID-19 vaccine, booster vaccine, third dose vaccine, omicron, heterologous primary vaccination, CoronaVac, ChAdOx1, cancer, immunogenicity, SARS-CoV-2

## Abstract

No data regarding the efficacy of a third mRNA vaccine for solid cancer patients previously primed with the heterologous CoronoVac/ChAdOx1 vaccination implemented in Thailand during the shortage of vaccine supply are available. Forty-four cancer patients who previously received the heterologous CoronaVac-ChAdOx1 regimen were boosted with a third mRNA COVID vaccine, either BNT162b2 or mRNA-1273. Anti-RBD IgG was measured immediately before, two weeks after, and four weeks after the third dose. The antibody response was compared to 87 age- and gender-matched cancer patients who were primed with the homologous ChAdOx1/ChAdOx1 regimens. Post-third dose anti-RBD IgG levels significantly increased compared to pre-third dose levels. There was no statistical difference in post-third dose antibody titers or neutralization levels between these two primary series regimens. Treatment with chemotherapy was associated with a lower antibody response compared to endocrine therapy/biologics. Similar antibody levels were observed after a third booster with either BNT162b2 or mRNA-1273 following heterologous CoronaVac/ChAdOx1 vaccination. There was no statistical difference in the immune response following the third-dose vaccination between cancer patients and healthy individuals who received the same heterologous CoronaVac/ChAdOx1 vaccination. In conclusion, a similar degree of enhanced immunogenicity was observed after a third mRNA COVID-19 vaccination in solid cancer patients who previously received the heterologous CoronaVac/ChAdOx1 regimens.

## 1. Introduction

Cancer patients are at elevated risk for severe Coronavirus disease 2019 (COVID-19) infection [1,2]. Diminished vaccine-induced immunogenicity and vaccine effectiveness following primary vaccination compared to the general population have been observed among cancer patients [3,4,5,6,7]. During the early phase of the COVID-19 pandemic, there was a limited vaccine supply and an uneven distribution of vaccines worldwide. The mRNA vaccines, including the BNT162b2 and mRNA-1273, were mainly available in the USA and Europe, while Asian countries including Thailand administered other vaccine platforms, including inactivated vaccines (e.g., Sinopharm-BBIBP and Sinovac-CoronaVac) and adenoviral-vectored vaccine (e.g., ChAdOx1).

To combat the limited and unpredictable vaccine supply during the SARS-CoV2 pandemic, heterologous primary schedules of COVID-19 vaccines were implemented in many countries. In Canada, Spain, and Germany, the heterologous ChAdOx1 followed by an mRNA vaccine demonstrated similar high vaccine efficacy compared to the homologous mRNA vaccine [8,9,10,11]. In Thailand, CoronaVac and ChAdOx1 vaccines were the two main COVID-19 vaccine platforms first available during the pandemic. The heterologous CoronaVac followed by ChAdOx1 vaccines was initiated by the Thai National Vaccine Committee as an alternative primary vaccination for healthy, non-elderly adults during the pandemic wave of Delta variant in July 2021, aiming to raise immunity in a shorter time period (a four- versus 8–10-week interval between first and second vaccines) compared to homologous ChAdOx1 vaccines [12,13]. As a consequence, a subgroup of mostly younger cancer patients also received an alternative heterologous regimen of heterologous CoronaVac primary series followed by ChAdOx1 vaccines.

Due to the emergence of the SARS-CoV-2 Omicron variant of concerns and the waning of vaccine-induced immune response, third dose vaccination with mRNA vaccine is now recommended [14,15,16,17,18]. In healthy adults, a third booster with mRNA COVID-19 vaccine following the heterologous CoronaVac/ChAdOx1 regimens demonstrated good vaccine efficacy against variants of concern [19]. No data regarding the efficacy of a third mRNA COVID-19 vaccine booster are available for solid cancer patients primed with this heterologous regimen. We aimed to assess the immunogenicity of a third booster following a primary series of the heterologous CoronoVac/ChAdOx1 vaccination in solid cancer patients.

## 2. Methods

### 2.1. Study Design and Participants

This study enrolled solid cancer patients aged ≥18 years who had been previously primed with the heterologous CoronaVac followed by ChAdOx1 vaccines, administered 4 weeks apart for at least 3 months, and had no history of SARS-CoV-2 infection. The recruitment period was between 27 December 2021, and 9 February 2022, at King Chulalongkorn Memorial Hospital, Bangkok, Thailand, and Phrapokklao Hospital, Chanthaburi, Thailand. Participants received a third mRNA COVID-19 vaccine, either BNT162b2 (Pfizer-BioNTech, Pearl River, NY, USA) or mRNA-1273 (Moderna, Norwood, MA, USA) according to vaccine availability. Blood samples were collected immediately before and 2–4 weeks after the third COVID-19 vaccination.

Demographics, including age, sex, and body mass index (BMI), and clinical data, including cancer type, current disease status, type of anticancer treatment (endocrine therapy without any previous chemotherapy versus chemotherapy treatment given during the COVID-19 immunization), and therapeutic corticosteroid use, defined as >10 mg prednisolone equivalent for more than 7 days, were collected and entered into the REDCap electronic data capture tools hosted at Chulalongkorn University [20,21]. Vaccine-related adverse events was recorded for seven days and severity was graded according to the FDA’s toxicity grading scale for healthy adult and adolescent volunteers [22].

All patients provided written, informed consent. The study was approved by the Institutional Review Board of Faculty of Medicine, Chulalongkorn University (No. 486/64) and the Chanthaburi Research Ethics Committee/Region 6 (CTIREC) (No. 044/64) and registered in the Thai Clinical Trials Registry (TCTR20220112004).

### 2.2. Study End-Points

The primary endpoint was the SARS-CoV2 antibody response after the third mRNA COVID vaccine in comparison with the standard homologous ChAdOx1/ChAdOx1 vaccines administered in solid cancer patients. The secondary end-point was to compare antibody response with healthy individuals who were primed with the same primary series of CoronaVac/ChAdOx1 vaccines. We defined anti-RBD IgG ≥ 300 BAU/mL as adequately protective. This value corresponded to the focus reduction neutralization titers (FRNT50) ≥ 40 against B.1.1.529 (Omicron) variant of concern BA.2 [23]. This anti-RBD IgG cut-off value had a high (92%) concordance rate with the presence of neutralizing antibodies against Omicron determined by ELISA-based surrogate neutralization assay [23].

### 2.3. Assessment of SARS-CoV-2 Binding Antibody and Neutralization against Omicron Variant

Serum samples were measured for anti-RBD IgG using commercial chemiluminescent immunoassays (AdviseDx SARS-CoV-2 IgG II, Abbott Diagnostics, Lake Forest, IL, USA) according to the manufacturer’s instructions. For anti-RBD IgG, measured units (AU/mL) were transformed into the WHO international standard unit (binding antibody unit; BAU/mL) using the equation: BAU/mL = 0.142x AU/mL. The value ≥ 7.1 BAU/mL (equal to 50 AU/mL) was considered positive.

Neutralization to Omicron BA.2 subvariant was performed using the cPass^TM^ SARS-CoV-2 neutralizing assay (GenScript Biotech, Piscataway, NJ, USA) [12]. Activity of neutralization was reported as percentage of inhibition and the cut-off values ≥ 30% inhibition were considered positive, indicating the presence of neutralizing antibodies according to the manufacturer’s instruction.

### 2.4. Comparison with the Homologous ChAdOx1/ ChAdOx1 Vaccines

In the same period, solid cancer patients previously primed with ChAdOx1/ChAdOx1 regimens were administered a third mRNA COVID-19 vaccine. Their immunogenicity data were reported [24]. Since patients who received the alterative CoronaVac/ChAdOx1 tended to be younger than those who received the standard ChAdOx1/ChAdOx1 regimen, we matched one patient primed with the CoronaVac/ChAdOx1 vaccines with two patients who were previously immunized with ChAdOx1/ChAdOx1 vaccines by age (±5 years) and sex. This resulted in eighty-seven solid cancer patients who were previously immunized with ChAdOx1/ChAdOx1vaccines as a reference vaccination regimen.

### 2.5. Comparison with Healthy Individuals

Data from healthy individuals came from a previously published study [19]. One hundred and seven healthy individuals who previously received the CoronaVac followed by the ChAdOx1 vaccine were administered either BNT162b2 or mRNA-1273 vaccines. Anti-RBD IgG levels measured immediately before and 2 weeks after the third vaccination were used for comparison.

### 2.6. Statistical Analysis

Anti-RBD IgG concentrations were reported as geometrical mean titers (GMT) with 95% confidence interval (CI). Pairwise comparisons of antibody concentration between the heterologous and homologous vaccination regimens or between cancer and healthy cohorts were performed with the nonparametric Mann–Whitney test. Comparison within the same patients was performed using the Wilcoxon matched pairs signed rank test. Chi-square or Fisher’s exact test was performed for comparisons of proportions.

Statistical analyses were performed using Stata 15 (Statacorp LLC, College Station, TX, USA) and GraphPad Prism version 9.0 (GraphPad Software, San Diego, CA, USA). All tests were two-sided with statistical significance set at *p*-value < 0.05.

## 3. Results

### 3.1. Post-Third Dose SARS-CoV2 Binding Antibody Concentration Comparison between the Primary Series of CoronaVac/ChAdOx1 and ChAdOx1/ChAdOx1 Regimens

Between 27 December 2021, and 9 February 2022, 44 solid cancer patients who previously received the heterologous CoronaVac/ChAdOx1 regimen were recruited for a third mRNA COVID-19 vaccine. In the same period, 87 age- and sex- matched solid cancer patients previously primed with the homologous ChAdOx1/ChAdOx1 regimen were selected as a reference group (Figure 1A). According to vaccination schedules, the median interval between the first and second dose for CoronaVac/ChAdOx1 was shorter than those of ChAdOx1/ChAdOx1 regimen (24 days [IQR 21–28] vs. 70 days [IQR 56–77]). The median time between the second and third was similar between these two vaccination regimens (127.5 days [IQR 113.5–137] versus 118 days [IQR 107–136]). Approximately 45% (20 out of 44) and 36% (31 out of 87) of the patients in the CoronaVac/ChAdOx1 and ChAdOx1/ChAdOx1 groups were boosted with BNT162b, respectively. Clinical data regarding cancer types, disease status, and anticancer treatment were similar between these two primary vaccination groups (Table 1).

The median age of patients was 57 years in both groups. Approximately 55% were female. Most patients were diagnosed with breast cancers, colorectal cancers, and cancer of the head and neck. Approximately 45% and 40% of patients had early-stage and recurrent or metastatic disease, respectively. Cancer treatment was categorized into two types including chemotherapy or hormonal therapy/biologics.

As expected, anti-RBD IgG concentrations at the pre-third dose time period were low and rose significantly after the third dose in both groups (CoronaVac/ChAdOx1: GMT 71.4 vs. 2330 BAU/mL, *p* < 0.0001 and ChAdOx1/ChAdOx1: GMT 40.6 vs. 2362 BAU/mL, *p* < 0.0001) (Figure 1B). There was also no statistical difference found between patients primed with CoronaVac/ChAdOx1 and ChAdOx1/ChAdOx1 schedules (GMT 71.4 vs. 40.6 BAU/mL, *p* = 0.1802). Following the third dose mRNA vaccine, levels of anti-RBD IgG in solid cancer patients primed with CoronaVac/ChAdOx1 significantly increased to comparably high titers to those receiving the primary series of ChAdOx1/ChAdOx1 regimen (GMT 2330 vs. 2362 BAU/mL, *p* = 0.84) (Figure 1B).

The proportion of patients with adequate response, defined by level above 300 BAU/mL, was significantly increased from 16% (seven out of 44) to 91% (40 out of 44) and from 10% (nine out of 87) to 90% (78 out of 87) in those who immunized with primary series of CoronaVac/ChAdOx1 and ChAdOx1/ChAdOx1 vaccines, respectively (*p* < 0.0001). Similar adequate immunity was observed after a third dose booster in solid cancer patients primed with either CoronaVac/ChAdOx1 or ChAdOx1/ChAdOx1 regimen.

### 3.2. Post-Third Dose SARS-CoV2 Binding Antibody Levels between Solid Cancer Patients and Healthy Individuals Who Primed with the CoronaVac/ChAdOx1 Vaccination

At pre-third dose, anti-RBD IgG of solid cancer patients were statistically lower than those of healthy individuals (GMT 71.4 vs. 136.7 BAU/mL, *p* = 0.004) (Figure 2A). This could be explained by the reduced immunogenicity in the cancer population as well as the younger age of healthy subjects compared to cancer patients (median age of 41 [IQR 35–48] versus 57 [IQR 48.5–65] years). Despite the difference in the pre-third dose levels, the antibody response after the third dose in cancer patients rose to a titer level that was not significantly different to healthy controls (GMT 2330 [95%CI 1487–3649] vs. 3823 [95%CI 3446–4241] BAU/mL, *p* = 0.34). An age-matched subset analysis between cancer and healthy adults was additionally performed to address the age discrepancy (Figure 2B). No statistically significant difference of either pre- or post-third dose antibody levels between cancer and healthy controls was observed in the age-matched subset analysis. When the cut-off value of 300 BAU/mL was applied for classification of adequate response, the proportion of adequate response markedly increased from 16% at pre-third dose to 91% (40 out of 44) for cancer patients and to 100% (107/107) for healthy patients at post-third dose. A small subgroup (9%) of cancer patients elicited an inadequate response. All these four poor responders were treated with a combination of chemotherapy, such as doxorubicin/cyclophosphamide and cisplatin/5-Fluorouracil (Appendix A). One patient had also received therapeutic steroids for congenital adrenal hyperplasia.

### 3.3. Impact of Anticancer Treatment and Type of mRNA COVID-19 on Post-Third Dose Antibody Response

Treatment with chemotherapy was associated with reduced antibody levels compared to hormonal therapy/biologics (GMT 1558 vs. 5069 BAU/mL, *p* = 0.01) (Figure 2C and Appendix A). Similar levels of anti-RBD IgG were observed when either BNT162b2 or mRNA-1273 was administered as the third COVID-19 vaccine (GMT 2037 vs. 2605 BAU/mL, *p* = 0.54).

### 3.4. Safety

Vaccine-related adverse events were assessed in 95% (42 out of 44) of cancer patients who received CoronaVac/ChAdOx1/mRNA and 98% (86 out of 87) of those received ChAdOx1/ChAdOx1/mRNA. No new serious adverse events were reported among patients who received a third dose vaccination following a primary series of the heterologous CoronaVac/ChAdOx1 vaccines at 2–4 weeks post-third vaccine. Approximately 48% (20 out of 42) of patients had any vaccine-related reactogenicity. Pain and tenderness were the most common local reactions, occurring in 48% and 36% of patients, respectively. Myalgia and fatigue were the most common systemic reactions, occurring in 26% and 19%, respectively (Figure 2D). Compared with the ChAdOx1/ChAdOx1/mRNA regimen, less tenderness (36% vs. 58%, *p* = 0.017) and fatigue (19% vs. 38% *p* = 0.028) occurred in patients vaccinated with CoronaVac/ChAdOx1/mRNA regimen. Lymphadenopathy, however, was more frequently observed in the CoronaVac/ChAdOx1/mRNA group (12% vs. 1% *p* = 0.014) (Appendix A).

### 3.5. Neutralization against Omicron Variant of Concern

Neutralization against the Omicron BA.2, dominant strain during the study period was measured in post-third dose serum of 44 and 40 patients primed with CoronaVac/ChAdOx1 and ChAdOx1/ChAdOx1 vaccines, respectively (baseline demographics showed in Appendix A). In accordance with the SARS-CoV2 binding antibody levels, patients who received the heterologous CoronaVac/ChAdOx1 regimen had a comparable proportion of detectable neutralizing antibody against the Omicron variant after the third dose vaccination compared to those immunized with a primary series of the homologous ChAdOx1/ChAdOx1 regimen (86.4% vs. 77.5%, *p* = 0.289) (Figure 3).

## 4. Discussion

Data regarding the heterologous vaccinations in cancer patients who are at higher risk of suppressed immunity are limited. To the best of our knowledge, this is the first report of immunogenicity in response to a third dose mRNA COVID-19 vaccination following the heterologous primary CoronaVac/ChAdOx1 vaccine schedule in solid cancer patients. This vaccination regimen was well-tolerated and elicited a high SARS-CoV2 binding antibody concentration comparable to the standard homologous ChAdOx1/ChAdOx1 vaccination followed by a third mRNA COVID-19 vaccine. Similar neutralization against the Omicron BA.2 subvariant was also demonstrated.

The heterologous mRNA/ vector/mRNA vaccination strategy has been reported to successfully induce an antibody response in a small case series of patients with lymphoma who poorly responded to primary vaccination [25,26]. Higher humoral and cellular immune response of heterologous vector/mRNA vaccine compared to the homologous vector/vector or mRNA/mRNA was also found in solid organ transplant recipients [27]. The vector-vaccine induced a higher cellular immune response than the mRNA-vaccine. In contrast, greater antibody response and neutralizing activity were observed with the mRNA-vaccine compared to vector vaccines [10,27]. Combining the advantages of both vaccine platforms could explain the enhanced immunity of the heterologous vaccine regimen.

Although reduced immunogenicity after two doses of COVID-19 vaccine has been demonstrated in cancer patients compared to the general population [5,6,28,29], a third dose booster with mRNA COVID-19 vaccine has been shown to raise the immunity of cancer patients to a level comparable to healthy individuals, as shown in the current study, and our previous study reported the comparable immunogenicity of the ChAdOx1/ChAdOx1/mRNA vaccination between cancer patients and healthy adults [24]. Despite the known inferior efficacy of inactivated or vector vaccine when compared to mRNA vaccines [30,31,32,33], third dose heterologous mRNA vaccination enhanced the immunogenicity of the primary vaccination with inactivated or vector vaccines to a level comparable to three doses of mRNA vaccines in healthy individuals [34,35].

Limitations of the study include the lack of cellular immune response and long-term protection. The sample size is small, and this limited additional subset analysis. In Thailand, cancer patients are considered a vulnerable group and most received the standard two doses of ChAdOx1 vaccines. Only a small groups of cancer patients received the alternative CoronaVac/ChAdOx1 primary vaccination in the period of the delta wave due to limited vaccine supply.

## 5. Conclusions

This study provides evidence for the safety and efficacy of the heterologous mRNA COVID-19 boosting following the heterologous primary vaccination with CoronaVac/ChAdOx1 vaccine schedule in solid cancer patients.

## 6. Simple Summary

A third dose booster is currently recommended to fight against the emerging SARS-CoV-2 Omicron variant of concern and waning of immunity over time. No data exist regarding the efficacy of a third mRNA COVID-19 booster for solid cancer patients who previously received a primary series of the heterologous CoronoVac/ChAdOx1 vaccination. We assessed the safety and humoral immunity in patients with solid malignancy following the CoronoVac/ChAdOx1/mRNA compared to the standard ChAdOx1/ChAdOx1/mRNA vaccination schedules and healthy individuals. The results of our study provide supportive evidence of the safety and efficacy of the CoronoVac/ChAdOx1/mRNA vaccination schedule in solid cancer patients.

## Figures and Tables

**Figure 1 vaccines-10-01613-f001:**
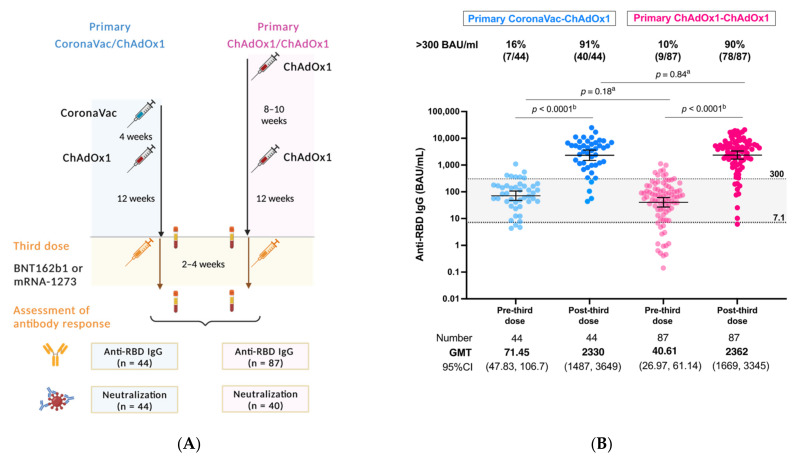
(**A**) Study flow diagram (**B**) SARS-CoV-2 binding antibody response at post-third mRNA COVID-19 vaccine following the primary heterologous CoronaVac/ChAdOx1 versus ChAdOx1/ChAdOx1 vaccination schedules. Anti-RBD IgG levels ≥ 7.1 BAU/mL (equal to 50 AU/mL) were considered positive, whereas levels >300 BAU/mL were considered as adequate response. ^a^ Mann-Whitney test, ^b^ Wilcoxon signed rank test.

**Figure 2 vaccines-10-01613-f002:**
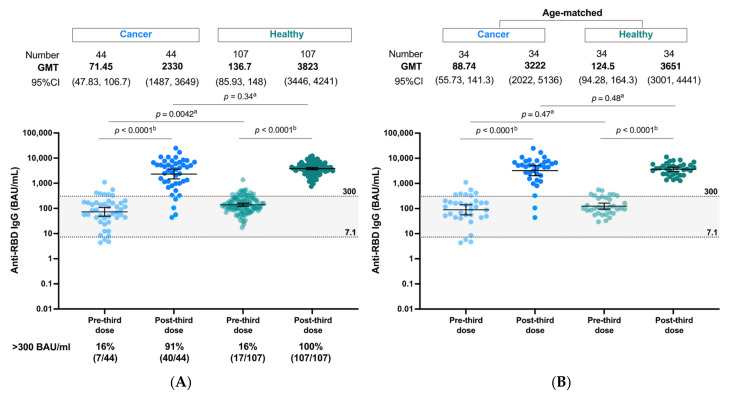
(**A**,**B**) SARS-CoV-2 binding antibody response SARS-CoV-2 binding antibody response at post-third mRNA COVID-19 vaccine following the heterologous CoronaVac/ChAdOx1 vaccination in cancer patients versus healthy controls (**A**) and aged match subset analysis (**B**). (**C**) SARS-CoV-2 binding antibody response at post-third mRNA COVID-19 vaccine following the primary CoronaVac/ChAdOx1 vaccination in cancer patients stratified by types of mRNA COVID-19 vaccines and types of anticancer treatment. (**D**) Vaccine-related reactogenicity after the CoronaVac/ChAdOx1/mRNA vaccination in cancer patients. ^a^ Mann-Whitney test, ^b^ Wilcoxon signed rank test.

**Figure 3 vaccines-10-01613-f003:**
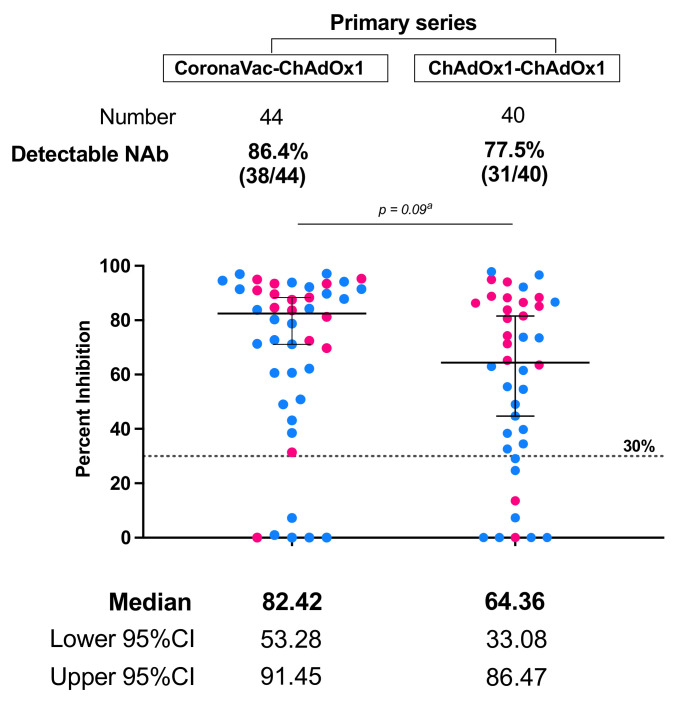
ELISA-based surrogate neutralization against Omicron BA2 variant in response to the CoronaVac/ChAdOx1/mRNA versus the ChAdOx1/ChAdOx1/mRNA vaccination in cancer patients. Data are reported as the median and 95%CI of percentage of inhibition between human ACE-2 and RBD protein. The cut-off value of 30% indicates the presence of detectable neutralization according to the manufacturer’s (cPass, Genscript) protocol. Data points represent individual samples. The pink color represents chemotherapy and the blue color represents treatment with hormonal therapy/biologics. ^a^ Mann-Whitney test.

**Table 1 vaccines-10-01613-t001:** Demographics and clinical characteristics.

	Cancer		Cancer		*p*-Value ^#^	Healthy		*p*-Value *
Primary Series of Vaccination	CoronaVac-ChAdOx1		ChAdOx1-ChAdOx1			CoronaVac-ChAdOx1		
	(n = 44)		(n = 87)			(n = 107)		
Age, years, median (IQR)	57	(48.5–65)	57	(48–65)	0.774	41	(35–48)	<0.001
Sex								
Female	24	(55%)	47	(54%)	0.955	46	(43%)	0.196
Male	20	(45%)	40	(46%)		61	(57%)	
BMI, kg/m2, median (IQR)	21.7	(19.5–25.5)	23.1	(21–26)	0.238			
Cancer types								
Breast	18	(41%)	36	(41%)	0.108			
Colorectal	11	(25%)	33	(38%)				
Head Neck	6	(14%)	4	(5%)				
Hepato-Biliary-Pancreatic	4	(9%)	5	(6%)				
Esophagus/Gastric	3	(7%)	1	(1%)				
Genitourinary	2	(5%)	2	(2%)				
Lung	0	(0%)	4	(5%)				
Other	0	(0%)	2	(2%)				
Cancer treatment								
Chemotherapy	29	(66%)	68	(78%)	0.131			
Hormonal therapy/Biologics	15	(34%)	19	(22%)				
Corticosteroid								
No/pre-medication	41	(93%)	86	(99%)	0.110			
Therapeutic purpose	3	(7%)	1	(1%)				
Disease status								
Early	20	(45%)	39	(45%)	0.695			
Locally advanced	8	(18%)	11	(13%)				
De novo metastasis	10	(23%)	27	(31%)				
Recurrence	6	(14%)	10	(11%)				
Co-morbidity								
Diabetes	6	(14%)	11	(13%)	0.873			
Hypertension	12	(27%)	18	(21%)	0.397			
Cardiovascular disease	3	(7%)	2	(2%)	0.334			
Respiratory tract disease	1	(2%)	2	(2%)	1.000			
Interval between first to second vaccine, days	24	(21–28)	70	(56–77)	<0.001	27	(21–28)	0.153
Interval between second to third vaccine, days	127.5	(113.5–137)	118	(107–136)	0.306	131	(106–138)	0.854
Interval between third dose to blood collection, days	14	(14–14)	14	(14–14)	0.211	14	(14–14)	0.544
Type of third vaccine								
BNT162b2 (Pfizer)	20	(45%)	31	(36%)	0.276	55	(51%)	0.507
mRNA-1273 (Moderna)	24	(55%)	56	(64%)		52	(49%)	

**^#^** Comparison between the primary CoronaVac/ChAdOx1 and ChAdOx1/ChAdOx1 regimen in cancer patients. * Comparison between healthy and cancer patients who received the primary heterologous CoronaVac/ChAdOx1 regimen.

## Data Availability

Data are available upon reasonable request to the corresponding author.

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
