# Peer review of "Immunogenicity after a Third COVID-19 mRNA Booster in Solid Cancer Patients Who Previously Received the Primary Heterologous CoronaVac/ChAdOx1 Vaccine"

_vaccines, 2022, doi:10.3390/vaccines10101613_

Round 1

Reviewer 1 Report

Reviewer’s Comments – Manuscript entitled “Immunogenicity after a third COVIS-19 mRNA booster in solid cancer patients previously received the primary heterologous CoronaVac/ChAdOx1 vaccine” by Luangdilok et al. (Vaccines)

 In the manuscript submitted by Luangdilok and co-workers, the Authors measured the anti-RBD Ig response and neutralizing antibodies in cancer patients who were vaccinated with the heterologous CoronaVac/ChAdOx1 vaccines and boosted (third dose) with an mRNA vaccine (BNT162b2 or mRNA-1273). Immune responses were measured at the time of the mRNA boost and 2 weeks after the boost and were compared with immune responses at the same time points of cancer patients who were vaccinated with the homologous ChAdOx1/ChAdOx1 regimen and boosted with an mRNA vaccine (BNT162b2 or mRNA-1273) and with healthy subjects vaccinated with the heterologous CoronaVac/ChAdOx1 vaccines and boosted with an mRNA vaccine. The results indicate that the third dose significantly increases the IgG and neutralization titers and that there was no significant difference between the two priming regimens (heterologous and homologous). In addition, there was no difference between the cancer patients and healthy subjects vaccinated with the same heterologous CoronaVac/ChAdOx1 vaccine plus mRNA vaccine regimen. No differences were also observed in antibody levels after the third dose with BNT162b2 or mRNA-1273 following the heterologous CoronaVac/ChAdOx1 vaccination.

The study addresses an important issue in the field of Sars-COV-2 infection with possible beneficial impact on vaccination programs for cancer patients. The results fit the scope of the journal and may be of interest to the readership of Vaccines.

The study however, from my point of view, misses some data and the comparison among groups are not complete.

Specific comments

-       The healthy control group should include older subjects in order to be age-matched with the cancer groups. In addition, a healthy age-matched control group vaccinated with the homologous ChAdOx1/ChAdOx1 vaccine is missing as proper control group of the cancer group receiving the same homologous regimen. It is not clear to me why this control was not included in the study and why the study is mainly focused on the heterologous priming regimen. Is the heterologous priming regimen the only choice and the homologous priming regimen interrupted? In any case the proper control group could be included in the study and in the different comparison shown in figures 2 and 3.

-       In figure 2 the panel letter C and D are missing. Also it would be interesting to see the safety data of the healthy subjects

-       In figure 3, is p=0.09 or 0.289 as reported in the text at line 237? In the ChAdOx1/ChAdOx1 vaccine regimen group the number of patients was 87, but in figure 3 the results of only 40 subjects are reported. Could you please clarify why and how these 40 subjects were chosen. How are the neutralization results compared to those of the healthy controls? It would be interesting to know.

-       Lines 268: they mention “our previously research (manuscript in submission)”…perhaps you may summarize it.

Minor comment

-       Some spelling mistakes

Reviewer 2 Report

Dr. Luangdilok, et al. reported the immunogenicity after a third booster shot in solid cancer patients previously who received CoronaVac/ChAdOx1 or ChAdOx1 vaccines.

Although it is very interesting for the readers of “Vaccines”, there are some points to be revised.

Major points

Are there any different efficacies or adverse events between patients who received BNT162b2 and who received mRNA-1273 in each cohort? 

Minor points

Title has a grammatical error: “who” is required after “patients”.

P. 2, L. 93, 95 

“The primary end-point” might be better instead of “The first end-point”.  Also, “The secondary end-point” might be better instead of “The second end-point”.

P. 3, L. 113, P. 9, L. 268 

Can we see the data regarding efficacy of ChAdOx1/ChAdOx1 regimen somewhere? The authors said “the data were published (manuscript in submission). Were the data published as the abstract of the certain congress? The data are also very important.  Author should briefly state the results of two-dose vaccination.  Could the patients with inadequate response after two-dose vaccination achieve seroconversion after a third booster vaccination? Could you clarify?

P. 5 Figure 1A, B

  “Heterologous CoronaVac/ChAdOx1” and “Primary CoronaVac-ChAdOx1” are the same meaning. Also, “Homologous ChAdOx1/ChAdOx1” are the same meaning. I think the phrase written in the same color should be the same labels for easy understanding.

P. 6, L.1 

  “SAR-CoV2” should be “SARS-CoV2”

P.6, L. 194

  When did the patients who were treated with chemotherapy receive a third vaccination? Just before the chemotherapy? 

 P. 6, L. 211, Figure 2D

  How did you decide the grade (mild, moderate and severe)of the adverse events? According to the CTCAE?

P. 7 

 Please add “Figure 2” and “C” in the appropriate place in this page. 

P. 8

 Please add “D” in the appropriate place.

 “primart series” should be “Primary”. 

Author Response

Author's Reply to the Review Report (Reviewer 2)

Dr. Luangdilok, et al. reported the immunogenicity after a third booster shot in solid cancer patients previously who received CoronaVac/ChAdOx1 or ChAdOx1 vaccines.

Although it is very interesting for the readers of “Vaccines”, there are some points to be revised.

Major points

1) Are there any different efficacies or adverse events between patients who received BNT162b2 and who received mRNA-1273 in each cohort? 

Answer: Thank you for reviewer’s comment.

Of all 44 cancer patients included in the heterologous CoronaVac/ChAdOx1/mRNA, there was 20 and 24 patients received BNT162b and mRNA-1273. Due to the low number of patients in each group, there is not enough power to test the difference of either efficacy or adverse events and we stated this limitation in the discussion section (Page 10 line 275).

However, we did an analysis about the efficacy between CoronaVac/ChAdOx1/BNT162b and CoronaVac/ChAdOx1/BNT162b in Figure 2C (Page 8) and also briefly describe the results on Page 6 line 206-207 as follows.

“Similar levels of anti-RBD IgG were observed when either BNT162b2 or mRNA-1273 was administered as the third COVID-19 vaccines (GMT 2037 vs 2605 BAU/ml, p =0.54).”

With regards to adverse events, no difference in vaccine-induced reactogenicity between BNT162b2 and mRNA-1273 was observed.

Minor points

  • Title has a grammatical error: “who” is required after “patients”.

Answer: Thank you for reviewer’s comment. This error had been corrected. (Page 1, Line 3)

P. 2, L. 93, 95 

“The primary end-point” might be better instead of “The first end-point”.  Also, “The secondary end-point” might be better instead of “The second end-point”.

Answer: Thank you for reviewer’s comment. This error had been corrected. (Page 2, Line 93, 95)

P. 3, L. 113, P. 9, L. 268 

Can we see the data regarding efficacy of ChAdOx1/ChAdOx1 regimen somewhere? The authors said “the data were published (manuscript in submission). Were the data published as the abstract of the certain congress? The data are also very important.  Author should briefly state the results of two-dose vaccination.  

Answer: Thank you reviewer for pointing this. We added a URL link of pre-print (below) in the manuscript and also briefly summarize the result of efficacy of ChAdOx1/ChAdOx1 regimen in 284 cancer patients versus healthy cohort was briefly mentioned the result Page 9 Line 267-268.

Reference

Luangdilok, Sutima and Wanchaijiraboon, Passakorn and Pakvisal, Nussara and Susiriwatananont, Thiti and Zungsontiporn, Nicha and Sriuranpong, Virote and Namkanisorn, Teerayuth and Sainamthip, Panot and Suntronwong, Nungruthai and Vichaiwattana, Preeyaporn and Kerr, Stephen J. and Wanlapakorn, Nasamon and Poovorawan, Yong and Teeyapun, Nattaya and Tanasanvimon, Suebpong, Immunogenicity and Omicron Neutralization Following a Third COVID-19 Vaccination in Solid Cancer Patients Previously Primed with Two Doses of Chadox1 Vaccine: A Prospective Cohort Study. Available at SSRN: https://ssrn.com/abstract=4216696

Could the patients with inadequate response after two-dose vaccination achieve seroconversion after a third booster vaccination? Could you clarify?

From Figure 1B, none of the patients in the Coronavac/ChAdOx1 regimen was seronegative after two-dose vaccination. So we instead us the adequacy cut-off at 300 BAU/ml which is corresponded to the focus reduction neutralization titers (FRNT50  ≥ 40 against B.1.1.529 (Omicron) variant of concern BA.2 (Suntronwong, N.; Assawakosri, S.; Kanokudom, S.; Yorsaeng, R.; Auphimai, C.; Thongmee, T.; Vichaiwattana, P.; Duangchinda, T.; Chantima, W.; Pakchotanon, P.; et al. Strong Correlations between the Binding Antibodies against Wild-Type and Neutralizing Antibodies against Omicron BA.1 and BA.2 Variants of SARS-CoV-2 in Individuals Following Booster (Third-Dose) Vaccination. Diagnostics 2022, 12, 1781.). This was mentioned in the method section Page 3 line 99. On the other hand, 14 of total 87 patients in the ChAdOx1/ChAdOx1 was seronegative after the second ChAdOx1 dose  and all become seropositive after the third vaccination.

We also added the cut-off in Figure 1B for clarification the cut-off values:

Anti-RBD IgG levels ≥7.1 BAU/ml (equal to 50 AU/ml) were considered positive according to manufacturer’s manual, whereas levels >300 BAU/ml were considered as adequate response.

P. 5 Figure 1A, B

  “Heterologous CoronaVac/ChAdOx1” and “Primary CoronaVac-ChAdOx1” are the same meaning. Also, “Homologous ChAdOx1/ChAdOx1” are the same meaning. I think the phrase written in the same color should be the same labels for easy understanding.

Answer: Thank you reviewer for pointing this. The Figure 1A, B (Page 6) has been re-phrase according to the reviewer suggestion.

P. 6, L.1 

  “SAR-CoV2” should be “SARS-CoV2”

Answer: Thank you for reviewer’s comment. This error had been corrected.

P.6, L. 194

  When did the patients who were treated with chemotherapy receive a third vaccination? Just before the chemotherapy? 

Answer: We recorded the chemotherapy given within 12 weeks before the third vaccination

We also clarified this point in the method section Page 2 line 84.

P. 6, L. 211, Figure 2D

How did you decide the grade (mild, moderate and severe) of the adverse events? According to the CTCAE?

Answer:  Thank you for reviewer’s comment. The severity according to FDA’s toxicity grading scale for healthy adult and adolescent volunteers (below). This was also added in the method section Page 2 line 87-89.

Reference

FDA, U.S. Toxicity Grading Scale for Healthy Adult and Adolescent Volunteers Enrolled in Preventive Vaccine Clinical Trials, Guidance for Industry. Available online: https://www.fda.gov/regulatory-information/search-fda-guidance-documents/toxicity-grading-scale-healthy-adult-and-adolescent-volunteers-enrolled-preventive-vaccine-clinical (accessed on Jan 8).

P. 7 

 Please add “Figure 2” and “C” in the appropriate place in this page. 

Answer: Thank you for reviewer’s comment. This error had been corrected.

P. 8

 Please add “D” in the appropriate place.

 “primart series” should be “Primary”. 

Answer: Thank you for reviewer’s comment. This error had been corrected.

Round 2

Reviewer 1 Report

The revised version is more suitable for publication

Author Response

The revised version is more suitable for publication

Thank you very much.

Reviewer 2 Report

P. 3, L. 95 and 97

  Please use “primary endpoint” and “secondary endpoint”.

  They have not been corrected.

The authors have shown “the chemotherapy given within 12 weeks before the third vaccination.”  Could you provide the data about the median interval (also ranges) between the last chemotherapy and the third vaccination?  The long interval might cause good results.

The figures have become better to understand after adding the cut-off lines. But I think not 0.71 but 7.1 is correct.  

Author Response

1) Page 3, L. 95 and 97

  Please use “primary endpoint” and “secondary endpoint”.

  They have not been corrected.

Answer: sorry for this error. We have corrected this on Page 3 Line 95 and 97.

2) The authors have shown “the chemotherapy given within 12 weeks before the third vaccination.”  Could you provide the data about the median interval (also ranges) between the last chemotherapy and the third vaccination?  The long interval might cause good results.

Answer: Thank you the reviewer for the comment. Unfortunately, we did not record the detail date of every chemotherapy cycle so we cannot provide the median interval.

Among 29 patients in primary CoronaVac/ChAdOx series who received chemotherapy, 28 patients were actively received chemotherapy and only 1 patient stop chemotherapy last 3 month ago. On the other hand, among 68 patients in primary ChAdOx/ChAdOx series, 29 (42%) patients still received chemotherapy during the third vaccination and 39 (57%) had stop chemotherapy with the median interval of 84 days (range 34-160 days). 

Actually, we mainly defined the criteria for endocrine therapy group. In endocrine group, patients were never administered with any chemotherapy. While in the chemotherapy group, patients had received or receiving chemotherapyduring the COVID-19 immunization.

So we have clarified the above criteria in the method on page 2 line 284-286.

3) The figures have become better to understand after adding the cut-off lines. But I think not 0.71 but 7.1 is correct.  

Answer: Thank you for your comments, we have corrected the cut-off in all the figures including Figure 1B, 2A, 2B and 2C.